# Ursolic Acid Alleviates Cancer Cachexia and Prevents Muscle Wasting via Activating SIRT1

**DOI:** 10.3390/cancers15082378

**Published:** 2023-04-20

**Authors:** Weili Tao, Ze Ouyang, Zhiqi Liao, Lu Li, Yujie Zhang, Jiali Gao, Li Ma, Shiying Yu

**Affiliations:** 1Department of Oncology, Tongji Medical College, Tongji Hospital, Huazhong University of Science and Technology, Wuhan 430030, China; 2Reproductive Medicine Center, Tongji Medical College, Tongji Hospital, Huazhong University of Science and Technology, Wuhan 430030, China; 3Division of Respiratory and Critical Care Medicine, Department of Internal Medicine, Tongji Medical College, Tongji Hospital, Huazhong University of Science and Technology, Wuhan 430030, China

**Keywords:** ursolic acid, cancer cachexia, skeletal muscle, SIRT1, Ex-527, NF-κB

## Abstract

**Simple Summary:**

Cancer cachexia is a multifactorial condition characterized by body weight loss and skeletal muscle wasting that negatively impacts the quality of life and survival of cancer patients. In this paper, we found that ursolic acid (UA) alleviated cancer cachexia and prevented muscle wasting via activating SIRT1, and thence inhibiting phosphorylation levels of NF-κB and STAT3. Furthermore, UA remained effective in the advanced stages of cancer cachexia. This is significant because no approved drug for the treatment of cancer-related muscle atrophy is available. UA may be a potential drug for cancer cachexia and has the possibility of translational application to clinical treatment.

**Abstract:**

Skeletal muscle wasting is the most remarkable phenotypic feature of cancer cachexia that increases the risk of morbidity and mortality. However, there are currently no effective drugs against cancer cachexia. Ursolic acid (UA) is a lipophilic pentacyclic triterpene that has been reported to alleviate muscle atrophy and reduce muscle decomposition in some disease models. This study aimed to explore the role and mechanisms of UA treatment in cancer cachexia. We found that UA attenuated Lewis lung carcinoma (LLC)-conditioned medium-induced C2C12 myotube atrophy and muscle wasting of LLC tumor-bearing mice. Moreover, UA dose-dependently activated SIRT1 and downregulated MuRF1 and Atrogin-1. Molecular docking results revealed a good binding effect on UA and SIRT1 protein. UA rescued vital features wasting without impacting tumor growth, suppressed the elevated spleen weight, and downregulated serum concentrations of inflammatory cytokines in vivo. The above phenomena can be attenuated by Ex-527, an inhibitor of SIRT1. Furthermore, UA remained protective against cancer cachexia in the advanced stage of tumor growth. The results revealed that UA exerts an anti-cachexia effect via activating SIRT1, thereby downregulating the phosphorylation levels of NF-κB and STAT3. UA might be a potential drug against cancer cachexia.

## 1. Introduction

Cancer cachexia is a multifactorial syndrome characterized by persistent loss of skeletal muscle, with or without loss of fat mass, which is not completely reversed by conventional nutritional support and leads to progressive functional impairment [1]. Cancer cachexia is estimated to contribute to 50–80% of all cancers and around 20% of all cancer-related deaths [2]. Skeletal muscle depletion is the most prominent characteristic of cachexia, associated with decreased treatment tolerance, significant quality of life deterioration, and poor prognosis [3,4]. Cancer-related muscle atrophy can be managed effectively with an approved drug, but there is no such drug available [4].

Systemic inflammation and oxidative stress play important roles in skeletal muscle atrophy [5,6]. Many pro-inflammatory cytokines, such as interleukin-6 (IL-6), transforming growth factor-β (TGFβ), interleukin-1β (IL-1β), and tumor necrosis factor-α (TNFα), have been established to promote muscle fiber breakdown [7,8]. Multiple signaling pathways are involved in this process, including signal transducer and activator of transcription 3 (STAT3) [9], nuclear factor-kappa B (NF-κB) [10], and p38 mitogen-activated protein kinase (MAPK) [11,12]. Exposure to a circulating inflammatory cocktail secreted by cancer cells induces hyperoxidation of fatty acids in muscle cells, leading to oxidative stress and impaired myotube growth [13]. Metabolic dysfunction is another essential mechanism of skeletal muscle wasting, characterized by increased breakdown and decreased synthesis due primarily to hyperactivation of the ubiquitin–proteasome and autophagy–lysosomal systems [14,15]. Transcriptional upregulation of genes encoding several E3 ligases, including muscle-specific RING finger protein 1 (MuRF1, or TRIM63) and muscle atrophy F-box only protein 32 (FBXO32, or Atrogin-1), is an idiosyncrasy of muscle atrophy [16]. The activation of these genes occurs through the transcription factors such as FOXO1 and FOXO3 in the Forkhead box O (FOXO) family [17]. Acetylation and deacetylation regulate FOXO1 and FOXO3 activity, with SIRT1 mediating the latter [18,19]. Muscle wasting can be alleviated to some extent by intervening in these factors or signaling pathways.

SIRT1, an NAD+-dependent protein deacetylase, is the best studied of the seven members of the mammalian sirtuin family and has been shown to regulate a large number of biological processes [20], such as energetic homeostasis [21], inflammation, oxidative stress [22], mitochondrial biogenesis [23], and autophagy [24]. Several tissue types, perhaps all, benefit from SIRT1 in various disease models. SIRT1 has recently been given increasing attention in muscle function [25], muscle physiology [26,27], balancing muscle cell differentiation [28], and proliferation [29,30], especially in cancer cachexia-induced muscle atrophy [31,32]. SIRT1 plays an essential role in inhibiting the production of inflammatory factors such as IL-6 and the NF-κB signaling pathway in colon and pancreatic cancer [31,32,33]. Furthermore, several studies have found that the content of SIRT1 decreased significantly in the course of cancer cachexia [31,34]. As mentioned above, cancer cachexia may be prevented and treated by targeting SIRT1. Small molecules or natural products that modulate the functions of SIRT1 are potential therapeutic agents.

Ursolic acid (UA, Figure 1A) is a pentacyclic triterpene natural compound found in the leaves, flowers, and fruits of medicinal herbs [35,36]. It possesses a wide range of biological functions, such as anti-inflammatory actions [37], antioxidant effects [38], anti-carcinogenic effects [39,40], thermogenesis activation [41], and anti-obesity effects [42,43]. It is worth noting that UA can alleviate muscle atrophy and reduce muscle decomposition in some disease models, such as fasting-induced muscle atrophy, denervation-induced muscle atrophy [44], chronic kidney disease model [45], and obesity-related illness model [41]. Therefore, we proposed UA might be a potential candidate for cancer cachexia. However, the role and molecular mechanism of UA in cancer cachexia are not clear yet.

In this study, we explored the role and mechanisms of UA treatment in cancer cachexia. We found that UA is effective at reversing Lewis lung carcinoma tumor cell (LLC)-conditioned medium-induced myotube atrophy in vitro and ameliorating LLC-induced animal cachexia and muscle atrophy in vivo. Mechanistically, UA inhibited inflammatory responses, decreased the concentration of cytokines in serum, and alleviated symptoms of cancer cachexia via activating SIRT1, thence downregulating the ubiquitin E3 ligase expression levels and the phosphorylation of NF-κB and STAT3. An inhibitor of SIRT1 (Ex-527) could restore the above phenomena triggered by UA, indicating that the therapeutic effect of UA was mediated by SIRT1 activation. Our studies provide insights into potential treatment strategies for cancer cachexia and suggest that UA might be a promising new approach against cancer cachexia.

## 2. Materials and Methods

### 2.1. Reagents and Antibodies

UA and Ex-527 were acquired from MedChemExpress (Princeton, NJ, USA). The primary antibody against myosin heavy chain (MyHC, MAB4470) was acquired from R&D Systems (Minneapolis, MN, USA). The primary antibodies against STAT3 (#9139), p-STAT3 (Yyr705) (#9145), NF-κB (p65) (#8242), p-NF-κB (p-p65) (Ser536) (#3033), and SIRT1 (#9475) were acquired from Cell Signaling Technology (Beverly, MA, USA); FOXO3a (#10849-1-Ap), FOXO1 (#18592-1-Ap), NOX4 (#14347-1-Ap), and GAPDH (#10494-1-Ap) were acquired from Proteintech (Wuhan, China); and Atrogin-1 (ab168372) and MuRF1 (ab172479) were acquired from Abcam (Cambridge, MA, USA).

### 2.2. Cell Culture

Murine myoblasts (C2C12 cells) and LLC cells were purchased from Peking Union Medical College Cell Bank (Beijing, China). C2C12 and LLC cells were incubated in a 37 °C incubator with 5% CO_2_ in high-glucose DMEM medium (Hyclone, Logan, UT, USA). In DMEM medium, 10% fetal bovine serum (FBS, Gibco, New York, NY, USA) was added, as was 1% penicillin/streptomycin (Hyclone, Logan, UT, USA). When C2C12 myoblasts grew to 80–90%, myotube formation could begin to be induced. DMEM medium containing 2% horse serum (Gibco, New York, NY, USA) was used as a differentiation medium (DM) in place of the proliferation medium. The DM was transposed daily. On days 6–8, myotube differentiation was complete.

### 2.3. The Conditioned Medium Collection

LLC cells were seeded in 10 mm dishes and reached 90% confluence. The proliferation medium was replaced with a serum-free DMEM medium and cultured for 48 h continuously. LLC medium was purified by centrifugation at 4 °C for 20 min at 3000 rpm, and then a sterile filter of 0.22 mm was used to filter the sample. LLC-conditioned medium (LCM) was the filtered medium and kept at −80 °C or used immediately. LLC tumor-conditioned medium (TCM) was a 1:1 dilution of LCM and DMEM containing 2% horse serum containing cachexia factors that caused myotube atrophy.

### 2.4. Cell Viability Assay

To evaluate cell viability, the CCK-8 assay was used. In 96-well culture plates, C2C12 myoblasts were seeded at a density of 1 × 10^4^ cells/well and differentiated into myotubes. The myotubes were cultured with various concentrations of UA (0, 1, 2.5, 5, and 10 µM) at 37 °C for 24, 48, and 72 h. After incubation, 10 μL of CCK-8 solution was added to each well and incubated for another 2 h at 37 °C. Microplate readers (BioTek, Winooski, VT, USA) were used to measure absorbance at a wavelength of 450 nm.

### 2.5. Immunofluorescence Staining and Myofiber Size Measurement

In a 24-well plate, different concentrations of UA were used to treat C2C12 myotubes for 72 h. After being washed with PBS 3 times, C2C12 myotubes were fixed for 20 min with 4% paraformaldehyde, permeabilized for 30 min with 0.5% Triton X100, and blocked with 1% BSA (Servicebio, Wuhan, China) at room temperature. After that, the primary antibody against MyHC was incubated overnight at 4 °C, followed by the secondary antibody (Proteintech, Wuhan, China) at room temperature for 1 h. DAPI was then used to stain myotubes. Following this, a fluorescence microscope (Leica, Wetzlar, Hesse, Germany) was used to capture myotube images. Over 100 myotube diameters were measured with ImageJ software in at least 10 fields.

### 2.6. Animal Models and Treatments

All animal experiments complied with the ARRIVE guidelines and the National Institutes of Health Guide for the Care and Use of Laboratory Animals. They were approved by the Institutional Animal Care and Use Committee of the Tongji Medical College, Huazhong University of Science and Technology. Six-week-old healthy male C57BL/6 mice were raised in a specific-pathogen-free environment with a constant temperature and humidity with a dark/light cycle of 12:12 h. LLC cells (1 × 10^6^ cells/100 μL PBS) were inoculated subcutaneously on the right flanks of the mice. The endpoint for all animal studies was day 21 post-inoculation.

The first study explored the therapeutic effect of UA in cancer cachexia. There were four groups of all mice randomly divided (n = 10 per group): control group, LLC tumor-bearing group, LLC tumor-bearing + 100 mg/kg UA-treated group, and tumor-bearing + 200 mg/kg UA-treated group. The control group was injected with the same amount of PBS without LLC cells. After tumor inoculation on day 7, mice received different doses of UA treatment by oral gavage daily. The detailed protocol of the experiment is depicted in Figure 2A. UA dose selection was based on chronic kidney disease models (100 mg/kg/day) [45], normal aging (200 mg/kg/day) [46], and ulcerative colitis (200 mg/kg/day) [47]. The control group and the tumor-bearing group were orally administrated with an equal dose of solvent solution daily by gavage. Every two days, the body weight and tumor size were measured. The conversion relationship between tumor weight and volume is 0.52 × tumor length × tumor width^2^ [48]. Tumor-free body weight is body weight minus tumor weight. All mice were euthanized after two weeks of treatment. The blood was collected into tubes and centrifuged for the serum to measure the contents of TNF-α, IL-6, and IL-1β by ELISA assay. The tumors and other tissues, including the heart, liver, spleen, kidney, epididymal fat, soleus, extensor digitorum longus (EDL), and gastrocnemius muscle, were weighed. Some tissues were used immediately or frozen in liquid nitrogen or fixed in 4% paraformaldehyde.

The second study investigated the molecular mechanism of UA alleviating tumor-induced cachexia. All mice were randomly divided into five groups (n = 10 per group): control group, LLC tumor-bearing group, LLC tumor-bearing + 200 mg/kg UA-treated (i.g.) group, LLC tumor-bearing group + 10 mg/kg Ex-527-treated (i.p.) group, LLC tumor-bearing + 200 mg/kg UA-treated + 10 mg/kg Ex-527-treated group. The xenograft tumor models were used to select the doses of Ex-527, which is a SIRT1 inhibitor (10 mg/kg/day, i.p.) [49]. The control group and LLC tumor-bearing group were given the same amount of solvent. The mice were sacrificed after 14 days of treatment. The measurements and treatments were the same as in the first study.

The third study assessed the efficacy of UA in different stages against cancer cachexia. All mice were randomly divided into four groups (n = 10 per group): the control group, which received the same amount of solvent, and the tumor-bearing group which started receiving UA medication (200 mg/kg, i.g.) on days 6, 10, and 14 after LLC cell inoculation. All mice were sacrificed on day 21. As explained in the first study, the main characteristics and sample acquisition methods were prepared in the same way.

### 2.7. Histological Examination

In each group, three gastrocnemius muscles were selected at random, fixed in 4% paraformaldehyde, and embedded in paraffin. Staining with hematoxylin and eosin (H&E) solutions was performed on paraffin-embedded tissues by standard procedures. A Leica microscope was used to capture images of muscle sections. The cross-sectional areas of myofibers were quantified using ImageJ software (NIH, Bethesda, MD, USA). In each group, the cross-sectional area of the gastrocnemius muscle per mouse/muscle was approximately 50 fibers.

### 2.8. RNA Extraction and Quantitative Reverse Transcription PCR

Total RNA was isolated from C2C12 myotubes and muscle samples using RNAiso Plus (Takara, Otsu, Shiga, Japan). The quantity and quality of purified RNA were assessed using a NanoDrop 2000 Spectrophotometer (Thermo Fisher Scientific, Waltham, MA, USA) by measuring the light absorbance at 280 nm, 260 nm, and 230 nm and calculating the 260/280 (A260/280) and 260/230 (A260/230) ratios (all samples with the A260/280  =  1.98–2.1, A260/230  =  2.0–2.1). Then, 1% agarose gel electrophoresis was applied to check the RNA integrity, followed by visualization using Syngene G (AlphaMetrix Biotech, Melsungen, Hesse, Germany). One microgram of total RNA was reverse transcribed into cDNA using Hi Script II QRT SuperMix (Vazyme, Nanjing, China). ChamQ Universal SYBR qPCR Master Mix (Vazyme, Nanjing, China) was used to carry out the qRT-PCR according to the instructions. Quantitative detection of mRNA including SIRT1, NOX4, FOXO3a, FOXO1, MuRF1, and Atrogin-1 was performed using a StepOne Real-Time PCR System (ABI, Foster City, CA, USA). These genes, which were normalized to GAPDH and β-actin, were analyzed. The ΔΔCt method was used to analyze the relevant expression levels [50]. The primers and NCBI Reference Sequences for the detection of mRNA are as follows: SIRT1 (NCBI Reference Sequence: NC_000076.7), forward: GCTGACGACTTCGACGACG, reverse: TCGGTCAACAGGAGGTTGTCT. NOX4 (NCBI Reference Sequence: NC_000073.7), forward: TGCCTGCTCATTTGGCTGT, reverse: CCGGCACATAGGTAAAAGGATG.MuRF1 (NCBI Reference Sequence: NC_000070.7), forward: CCAGGCTGCGAATCCCTAC, reverse: ATTTTCTCGTCTTCGTGTTCCTT. Atrogin-1 (NCBI Reference Sequence: NC_000081.7): forward: CAGCTTCGTGAGCGACCTC, reverse: GGCAGTCGAGAAGTCCAGTC. GAPDH (NCBI Reference Sequence: NC_000072.7), forward: AGGTCGGTGTGAACGGATTTG, reverse: GGGGTCGTTGATGGCAACA. β-actin (NCBI Reference Sequence: NC_000071.7), forward: CTGTCCCTGTATGCCTCTG, reverse: ATGTCACGCACGATTTCC.

### 2.9. Western Blot Analysis

Total gastrocnemius muscle and C2C12 myotube protein were extracted on ice for 30 min with RIPA lysis buffer (Servicebio, Wuhan, China), which contained 1% phenylmethylsulfonyl fluoride (PMSF) and 1% Phosphatase Inhibitor Cocktail (Servicebio, Wuhan, China). Next, the supernatant proteins were collected after lysis by centrifugation at 12,000 rpm at 4 °C for 15 min. The concentration of protein was measured using BCA Protein Assay Kits (Beyotime, Shanghai, China). Equal amounts of protein were separated using 10% SDS-PAGE gel and transferred to a 0.45-µm polyvinylidene fluoride membrane (Millipore, Billerica, MA, USA). After 1 h of blocking with 5% skim milk at room temperature, the membrane was placed in primary antibodies and incubated at 4 °C overnight. Then, the membranes were incubated with secondary antibodies (Proteintech, Wuhan, China) at room temperature for 1 h after being washed 3 times each for 10 min with Tris-Buffer saline (TBST) containing 0.1% Tween-20. Finally, A three-time wash was followed by visualization with ECL solution (Thermo Fisher Scientific, Carlsbad, CA, USA). Syngene G (AlphaMetrix Biotech, Melsungen, Hesse, Germany) was used to capture images, and ImageJ software v1.8.0 was used to quantify each blot band.

### 2.10. ELISA Assay

The supernatant was collected by centrifugation of blood collected from mice at 3000 rpm at 4 °C for 15 min. According to the manufacturer’s instructions, levels of IL-6 and IL-1β were quantified using a commercial mouse ELISA kit acquired from Multsciences (Hangzhou, China), and the TNF-α mouse ELISA kit was obtained from Thermo Fisher Scientific (Carlsbad, CA, USA); the serum assay was repeated for each animal.

### 2.11. Molecular Docking

To understand how the SIRT1 protein and UA are bound, molecular docking experiments were conducted in AutoDock Vina 1.1.2. We downloaded the crystal structure of SIRT1 from the RCSB Protein Data Bank (www.rcsb.org (accessed on 13 May 2022)), and we downloaded the structure of UA from the PubChem database (https://pubchem.ncbi.nlm.nih.gov/ (accessed on 13 May 2022)). The exhaustiveness was set to 10, and the rest of the parameters were set by default. Before molecular docking, the protein structures were processed with AutoDocktools1.5.6. The interaction mode of the docking results was analyzed with PyMOL2.3.0. 

### 2.12. Statistical Analysis

Data and results are expressed as the mean ± SD, and statistical analysis was conducted using one-way ANOVA or Student’s *t*-test followed by Dunnett post hoc test for between-group significance. All experiments were repeated three times. Statistically significant differences were accepted at *p* < 0.05.

## 3. Results

### 3.1. UA Alleviates TCM-Induced Myotube Atrophy In Vitro

To investigate the anti-cachexia role of UA, C2C12 myotubes were cultured with TCM to obtain a cachexia model; we originally cultured C2C12 myotubes with different concentrations of UA (0.5–5.0 μM) for 24, 48, and 72 h. The results of the CCK8 assay showed that UA was not toxic to C2C12 myotubes at concentrations less than 5.0 μM (Figure 1B), so 1.0, 2.5, and 5.0 μM were chosen for subsequent experiments. UA (0.5−5.0 μM) dose-dependently reversed TCM-induced myotube atrophy, which was determined by morphological changes and myotube fiber width (Figure 1C). Furthermore, UA significantly suppressed TCM-induced MuRF1 and Atrogin-1 mRNA (Figure 1D,E) and protein expression (Figure 1F). The protein expression of MyHC was also increased by UA treatment in C2C12 myotubes (Figure 1F). The results showed that UA had a certain protective effect on TCM-induced myotube atrophy and reduced the degradation of muscle protein mediated by ubiquitin.

### 3.2. UA Alleviates Muscle Wasting and Prevents Cancer Cachexia In Vivo

Then, we evaluated the anti-cachexia effect of UA in LLC cachexia animals (Figure 2A). The tumor-free body weight of LLC tumor-bearing mice was significantly reduced (Figure 2B,C), and the weight of gastrocnemius muscle, tibialis anterior muscle (Figure 2E–G), epididymis fat (Figure 2J), heart, kidney, soleus muscle, EDL muscle, and liver also decreased significantly (Figure 2K,L and Appendix A) compared with the control group. In contrast, spleen weight increased significantly (Figure 2M). Compared with the LLC tumor group, UA dose-dependently significantly improved core features of cancer cachexia and significantly increased the tumor-free body weight and weights of the gastrocnemius, tibialis anterior, and epididymal fat mass (Figure 2B,C,E–G,J). In addition, UA was able to reduce spleen weight statistically significantly at the dose of 200 mg/kg, which might be related to UA’s ability to suppress inflammation (Figure 2M). Histological examination of gastrocnemius muscle revealed that UA significantly increased muscle fiber size in a dose-dependent manner (Figure 2H). In addition, Western blot showed that UA dose-dependently inhibited the protein expression of MuRF1 and Atrogin-1 in the gastrocnemius muscle (Figure 2I), whereas there was no inhibitory effect on the growth of LLC tumors at doses of 100 and 200 mg/kg (Figure 2D). These results corroborated the role of UA in preventing cancer cachexia and alleviating muscle wasting in vivo, which was not a consequence of reducing tumor burden.

### 3.3. UA Upregulates SIRT1 Expression and Inhibits the Expression of NOX4, FOXO1, and FOXO3a In Vitro and In Vivo

To explore the underlying mechanism of the anti-cachexia effect of UA, we examined the interaction pattern of SIRT1 protein with UA, as well as the influence of SIRT1 activation and muscle atrophy-related signaling pathway changes. Then molecular docking was performed to predict specific binding modes within the active site of SIRT1 by AutoDock Vina 1.1.2. The theoretical binding mode is shown in Figure 3A. The results demonstrated that UA bound to the ASM-147 domain of SIRT1 by forming a hydrogen bond, with bond lengths of 3.0 Å and 3.1 Å and a binding energy of −8.5 kcal/mol, which proved to have a good binding effect. In addition, UA formed hydrophobic interaction with the amino acid residues of LEU-197, LEU-198, ILE-219, GLU-222, PRO-223, HIS-150, THR-146, and LEU-144 of SIRT1. Further research on the mechanism of UA against cancer cachexia can be conducted based on the docking results.

The mRNA and protein expression of SIRT1 and NOX4 was significantly downregulated in the LLC tumor groups compared to the control group, as well as FOXO1 and FOXO3a in vitro. Meanwhile, UA could dose-dependently increase the expression of the aforementioned proteins (Figure 3B,C). Based on the results in the cell experiments, we also validated this result in gastrocnemius muscles of LLC tumor-bearing mice (Figure 3D).

### 3.4. UA Suppresses Phosphorylation of STAT3 and p65 Expression In Vitro and In Vivo and Reverses Elevated Serum Cytokines in the Cancer Cachectic Model

We examined the alterations of intracellular signaling pathways related to muscle wasting. UA remarkably downregulated the phosphorylation levels of STAT3 and p65 in vitro and in vivo as shown by Western blot (Figure 4A,B). Moreover, elevated levels of pro-inflammatory factors TNF-α (Figure 4C), IL-1β (Figure 4D), and IL-6 (Figure 4E) in the serum of LLC tumor-bearing mice could be detected by ELISA assay, whereas UA could obviously reduce the concentration of inflammatory factors in a dose-dependent manner.

### 3.5. UA Relieves TCM-Induced Myotube Wasting through SIRT1 Activation

We used the SIRT1 inhibitor Ex-527 (0.1 μM, the tumor cell model was used to select the dose [31]) to investigate whether SIRT1 activation mediated the therapeutic effect of UA on cancer cachexia and evaluated its effects on myotube atrophy in vitro. The Ex-527 treatment significantly reduced fiber width and aggravated myotube atrophy compared with the control group. UA treatment conspicuously reversed TCM-induced myotube atrophy, whereas the improvement was significantly attenuated by the SIRT1 inhibitor (Figure 5A). Meanwhile, the SIRT1 inhibitor also weakened the inhibitory effect of UA on the mRNA and protein expression of MuRF1 and Atrogin-1 in the C2C12 myotubes, as well as the enhancive effect of UA on the protein level of MyHC (Figure 5B,D). Moreover, Ex-527 treatment reduced the reinforcement effect of UA on SIRT1 at the protein and mRNA expression levels, while reducing the inhibitory effect of UA on NOX4, FOXO1, and FOXO3a protein expression levels (Figure 5C,E). Next, we explored alterations in signaling pathways in the C2C12 myotube atrophy model. The Western blot results showed that the remarkably downregulated phosphorylation levels of STAT3 and p65 by UA treatment could be elevated under the influence of the SIRT1 inhibitor (Figure 5F).

### 3.6. UA Improves Muscle Wasting and Prevents Cancer Cachexia in LLC Tumor-Bearing Mice through SIRT1 Activation

To delve into the role of SIRT1, we used Ex-527 and assessed its effects on muscle wasting and cancer cachexia in vivo. The detailed protocol of the experiment is depicted in Figure 6A. In LLC tumor-bearing mice, UA treatment significantly increased the tumor-free body weight, as well as the weight of gastrocnemius muscle, tibialis anterior muscle, soleus muscle, EDL muscle, and epididymal fat, whereas the improvement was significantly decreased by Ex-527 treatment (Figure 6C–F,I and Appendix A). However, tumor weight did not appear to be affected by them (Figure 6B). Furthermore, Ex-527 conspicuously attenuated the upregulation effect of UA on the protein levels of MuRF1 and Atrogin-1 in the gastrocnemius muscle of LLC tumor-bearing mice (Figure 6G). In addition, the histological examination of gastrocnemius muscle revealed that the SIRT1 inhibitor significantly decreased the improvement effect of UA on muscle fiber size (Figure 6H). The change in tumor-free body weight during the 21 days of the animal experiment is shown in Appendix A. The tumor-free body weight significantly increased by UA treatment was obviously reduced by Ex-527 treatment. However, the SIRT1 inhibitor attenuated the mitigating effect of UA on spleen mass gain (Appendix A). In addition, the weight of the liver, kidney, and heart was not significantly affected by Ex-527 in vivo (Appendix A–G).

Western blot showed that Ex-527 decreased the enhancive effect of UA on SIRT1 at the protein expression level while reducing the inhibitory effect of UA on NOX4, FOXO1, and FOXO3a protein levels (Figure 7A). Meanwhile, Ex-527 treatment distinctly increased the phosphorylation levels of STAT3 and p65 in gastrocnemius muscles (Figure 7B). Furthermore, we found that the levels of pro-inflammatory factors TNF-α, IL-1β, and IL-6 in the serum of LLC tumor-bearing mice lowered by UA could be raised by Ex-527 treatment (Figure 7C–E). In summary, UA appeared to have anti-cachexia effects through SIRT1 activation.

### 3.7. Effects of Delaying Treatment with UA until Advanced Stages of Tumor and Cachexia Progression in LLC Tumor-Bearing Mice

The diagnosis of early cancer cachexia has limitations due to the complex mechanism and multiple phenotypes of cachexia, so the early cancer cachexia treatment strategy has poor clinical practicability. In the preceding experiments, UA treatment was initiated at an early stage of the disease when no overt signs of wasting were detected. To investigate whether initiating UA treatment at a later time point could also prevent cancer cachexia, we initiated UA treatment in LLC tumor-bearing mice at 6, 10, and 14 days after tumor cell injection (Figure 8A).

On day 6, 10, or 14, LLC tumor/tumor-bearing mice received UA treatment, and body weight loss was limited to 6.0%, 10.1%, and 16.4% (Figure 8C), whilst the mice lost 27.7% of their tumor-free body weight without appreciable effect on tumor growth (Figure 8B). Meanwhile, UA preserved gastrocnemius weight and, to a lesser extent, tibialis anterior, soleus, and EDL weight from day 6 or 10 (Figure 8E,F and Appendix A). Moreover, UA relieved epididymal fat atrophy and spleen weight gain from day 6 or 10 (Figure 8D,G). However, unlike the effect of UA on tumor-free body weight, there were no significant effects on the liver, kidney, and heart mass in vivo, as shown in Appendix A–F. Furthermore, the levels of pro-inflammatory factors TNF-α, IL-1β, and IL-6 in the serum of LLC tumor-bearing mice were lowered after UA treatment from day 6, 10, or 14 (Figure 8H–J). Western blot showed that the expression of MuRF1 and Atrogin-1 in the gastrocnemius muscle decreased on day 6, 10, or 14 after UA treatment. However, with the delay of UA application, the inhibitory effect of UA on MuRF1 and Atrogin-1 weakened, with their levels increasing gradually (Figure 9A). Furthermore, UA upregulated SIRT1 expression from day 6, 10, or 14 and alleviated the elevation of NOX4, FOXO1, and FOXO3a protein levels from day 6 or 10 (Figure 9B). In addition, UA treatment downregulated the phosphorylation levels of p65 and STAT3 on day 6 or 10 (Figure 9C).

## 4. Discussion

Cancer cachexia is a multifactorial condition characterized by body weight loss and skeletal muscle wasting that negatively impacts the quality of life and survival of cancer patients [2,51]. Research in recent years provides better insights into molecular mechanisms and new therapeutic approaches to cancer cachexia [52,53]. Nevertheless, the therapeutic approaches or agents for muscle wasting of cancer cachexia are currently limited.

UA has shown potential anti-inflammatory [54], antioxidant [55], and antitumor effects [56], and it has shown positive effects in some muscular wasting models, such as chronic kidney disease models [45] and obesity-related disease models [41]. However, the effect of UA on cancer cachexia is highly unclear. To explore the role of UA on cancer cachexia, we initially evaluated the effect of UA in the C2C12 myotube atrophy model induced by LLC TCM. The results showed that UA was not toxic to normal C2C12 myotube within a concentration of 5.0 μM. Moreover, UA could dose-dependently alleviate myotube atrophy induced by LLC TCM and increase the diameter of the myotube, as shown by immunofluorescence staining. In addition, the expression of MuRF1 and Atrogin-1 at the mRNA and protein levels were decreased UA in a dose-dependent manner, and the protein level of MyHC was increased in the same way. Then, we further explored the effect of UA in LLC tumor-bearing mice, and results showed that UA could increase the mass of the tumor-free body, gastrocnemius muscle, tibial anterior muscle, and epididymal fat within a concentration of 200 mg/kg. Furthermore, HE showed that the cross-sectional area of myofibers was increased under UA treatment, and the expression of MuRF1 and Atrogin-1 was inhibited. Unfortunately, tumor weight was not significantly affected by UA treatment. The above results show that UA can significantly relieve the progression of cancer cachexia and prevent muscle wasting, which is not related to the antitumor effect. More importantly, in LLC tumor-bearing mice, the administration of UA during the advanced stage of tumor growth could still retard the muscle atrophy progression and prevent the occurrence of cachexia, which has important therapeutic significance.

UA has been demonstrated to increase the sensitivity of cancer cells to chemotherapy drugs by inhibiting NF-κB and enhancing the chemotherapeutic agent-induced cancer cell apoptosis [57]. Moreover, UA was shown to promote autophagy of cancer cells, leading to non-apoptotic cell death and delayed tumor growth in vivo [58]. These studies suggest the anticancer effects of UA. In addition, it has been reported that UA induces endoplasmic reticulum (ER) stress and autophagy in MCF-7 human breast cancer cells, and autophagy-dependent endoplasmic reticulum stress protects cells from UA-induced apoptosis through EIF2AK3-mediated MCL1 upregulation. UA may have a dose–response pattern, with UA causing a dose-dependent increase in MCL1 expression at concentrations of 15 and 20 μM, while at concentrations of 25 μM and above, UA becomes a negative regulator of MCL1 in MCF-7 cells and promotes apoptosis [59]. SIRT1 plays a dual role in cancer. According to the cellular environment, SIRT1 may act as both a tumor suppressor and a tumor promoter and may have different signaling targets in different cancer types [60,61]. In this study, it is possible that the anticancer and pro-cancer effects of UA and SIRT1 were balanced, as UA had no effect on the growth of LLC tumors, suggesting that UA may have a selective anticancer effect.

To explore the molecular mechanism by which UA alleviates cachexia and prevents muscle wasting, we performed molecular docking on UA and SIRT1 protein. The results revealed a good binding effect. The binding energy of UA to SIRT1 protein was −8.5 kcal/mol, and UA formed hydrogen bonds with amino acid residue ASM-147 with lengths of 3.0 Å and 3.1 Å. The regulatory effect of UA on SIRT1 was confirmed in vitro and in vivo. UA upregulated SIRT1 expression and downregulated the MuRF1 and Atrogin-1 expression in both cell and animal experiments, which is consistent with the report that treatment of skeletal muscle satellite cells with UA upregulates the expression of SIRT1 (~35-fold) significantly [46]. Meanwhile, UA reduced the expression levels of NOX4, FOXO1, and FOXO3a. The role of the SIRT1–NOX4 axis in cachexia has been confirmed [31]; UA may play an anti-cachexia role by regulating the SIRT1–NOX4 axis. In addition, UA downregulated NF-κB and STAT3 phosphorylation levels after activating SIRT1, and these two pathways play a key role in regulating cancer cachexia [4,62]. The remission effect of SIRT1 activation on cancer cachexia models could be reversed by the inhibitor of SIRT1, Ex-527. Therefore, the results show that UA alleviates cancer cachexia and prevents muscle wasting by activating SIRT1 and inhibiting NF-κB and STAT3. In addition, UA increases mitochondrial biogenesis and myoglobin expression in skeletal muscle and increases type IIA fibers, providing evidence that UA may enhance physical performance [46]. In future studies, we can further explore the effects of UA on muscle function and other physical performance factors in cancer cachexia.

Systemic inflammation is one of the important characteristics of cancer cachexia, and inflammatory cytokines IL-1β, IL-6, and TNF-α in the serum of cancer cachexia patients have been confirmed [63]. Elevated cytokines can activate NF-κB and STAT3 pathways. In turn, NF-κB and STAT3 pathway activation elevates the expression of cytokines, which aggravates cancer cachexia [63]. Consistent with the literature, we observed that UA downregulated elevated IL-6, IL-1β, and TNF-α levels in the serum of LLC tumor-bearing mice and inhibited elevated phosphorylation of NF-κB and STAT3. These effects were inhibited by Ex-527 but not be reversed completely. Interestingly, UA could dose-dependently weaken the increased spleen weight in LLC tumor-bearing mice at concentrations within 200 mg/kg, and the cytokines IL-6, IL-1β, and TNF-α are produced by immune cells or tumor cells. It has been reported that UA enhanced the autophagy of macrophages by upregulating the expression of Atg5 and Atg16L1. UA reduced IL-1β secretion in macrophages. In addition, serum IL-6 and TNF-α levels significantly decreased in UA-treated mice, which may be secondary to IL-1β reduction and the resulting reduced chronic systemic inflammation in mice [64]. Furthermore, UA inhibited activation, proliferation, and cytokine secretion in T cells, B cells, and macrophages. UA also significantly reduced the serum levels of IL-6 and IFN-γ [65]. Thus, we speculate that UA may affect immune cells, which may explain the effect of UA on cytokines that do not completely disappear after the application of Ex-527. UA may have effects on immune cells, tumor cells, and muscle cells in cancer cachexia, and their interactions need to be explored in more studies.

## 5. Conclusions

In conclusion, we found that UA alleviates cancer cachexia and prevents muscle wasting via activating SIRT1, and thence inhibiting phosphorylation levels of NF-κB and STAT3. Furthermore, UA remains effective in the advanced stages of cancer cachexia. UA may be a potential drug for cancer cachexia.

## Figures and Tables

**Figure 1 cancers-15-02378-f001:**
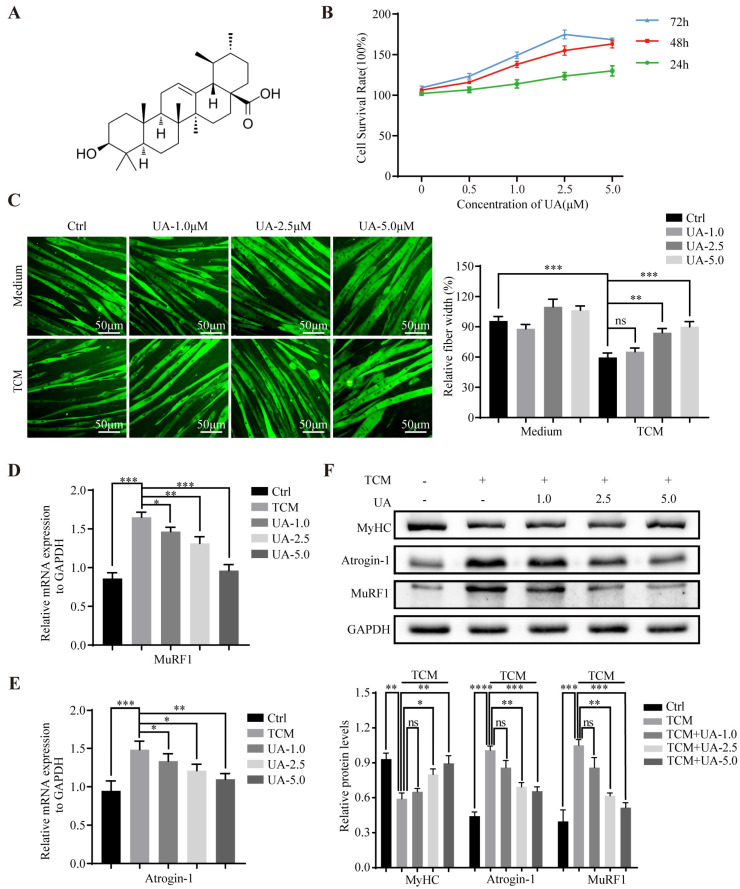
UA alleviates TCM-induced myotube atrophy in vitro. (**A**) The chemical structure of UA. (**B**) Effects of different concentrations of UA on the viability of C2C12 myotubes. C2C12 myotubes were cultured in different concentrations of UA (0, 0.5, 1.0, 2.5, 5.0 μM for 24, 48, and 72 h), and the viability of myotubes was detected by the CCK8 assay. Myotubes were cultured in various concentrations of UA under stimulation of TCM for 48 h. (**C**) Representative images of immunofluorescence staining for MyHC (green) are displayed (left) with different concentrations of UA. Scale bars = 50 μm. The relative fiber widths of each experiment were measured and calculated (right). mRNA levels of MuRF1 (**D**) and Atrogin-1 (**E**) in C2C12 myotubes were analyzed by real-time PCR and normalized to GAPDH. (**F**) Western blot was used to determine the expression of indicated proteins. The band intensities were quantified and normalized to GAPDH. * *p* < 0.05, ** *p* < 0.01, *** *p* < 0.001, **** *p* < 0.0001 versus control, ns: not significant, *n* = 3.

**Figure 2 cancers-15-02378-f002:**
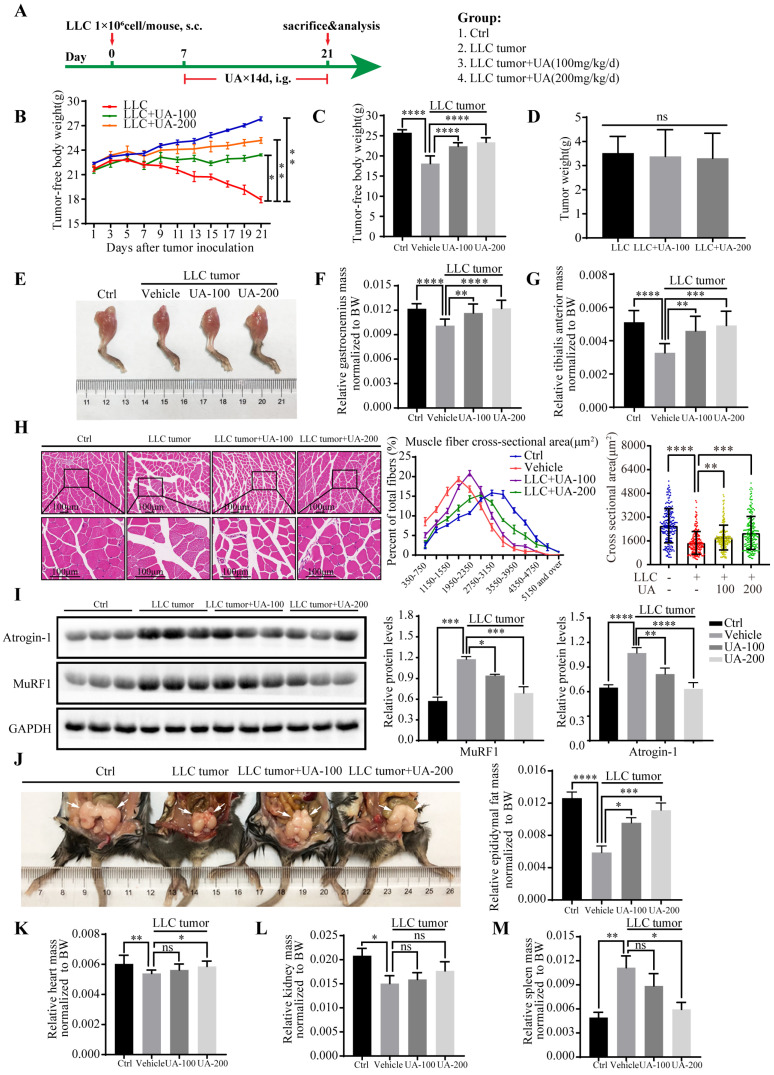
UA alleviates muscle wasting and prevents cancer cachexia in vivo. (**A**) Schematic illustration of the animal experimental design. The effects of UA on the main features of cachexia were examined, including (**B**,**C**) tumor-free body weight, (**D**) tumor weight, (**E**–**G**) gastrocnemius and tibialis anterior muscle mass, (**J**) epididymal fat mass, (**K**–**M**) heart, kidney, and spleen mass (BW: body weight). (**H**) Gastrocnemius muscle was observed histologically by H&E staining. Scale bars = 100 μm. The cross-sectional areas of approximately 210 myofibers per group were determined. *n* = 3 mice/group. (**I**) The protein expression of MuRF1 and Atrogin-1 in gastrocnemius muscles was detected with Western blot. The band intensities were quantified and normalized to GAPDH. * *p* < 0.05, ** *p* < 0.01, *** *p* < 0.001, **** *p* < 0.0001 versus control, ns: not significant, *n* = 10 mice/group.

**Figure 3 cancers-15-02378-f003:**
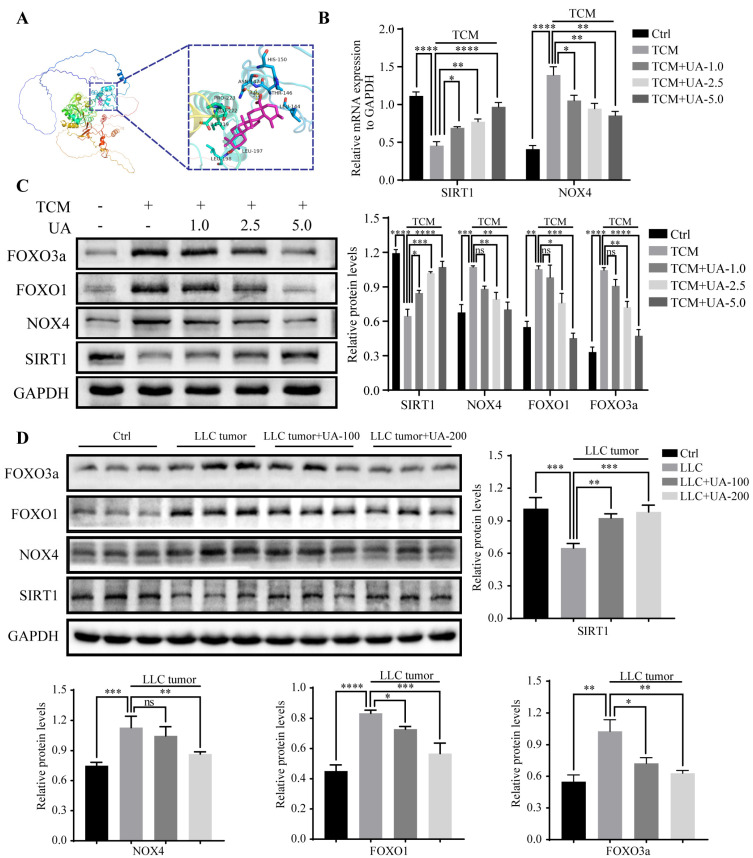
UA upregulates SIRT1 expression and inhibits the expression of NOX4, FOXO1, and FOXO3a in vitro and in vivo. (**A**) Molecular docking analysis of the interaction between UA and SIRT1 domain. (**B**) The mRNA expression of SIRT1 and NOX4 in the C2C12 myotube model was analyzed by real-time PCR and normalized to GAPDH. The expression levels of SIRT1, NOX4, FOXO1, and FOXO3a (**C**) in the C2C12 myotube model and (**D**) in the gastrocnemius muscles of the animal model were determined by Western blot. The band intensities were quantified by densitometry, and GAPDH was used as a control. * *p* < 0.05, ** *p* < 0.01, *** *p* < 0.001, **** *p* < 0.0001 versus control, ns: not significant, *n* = 3 in vitro and *n* = 10 mice/group in vivo.

**Figure 4 cancers-15-02378-f004:**
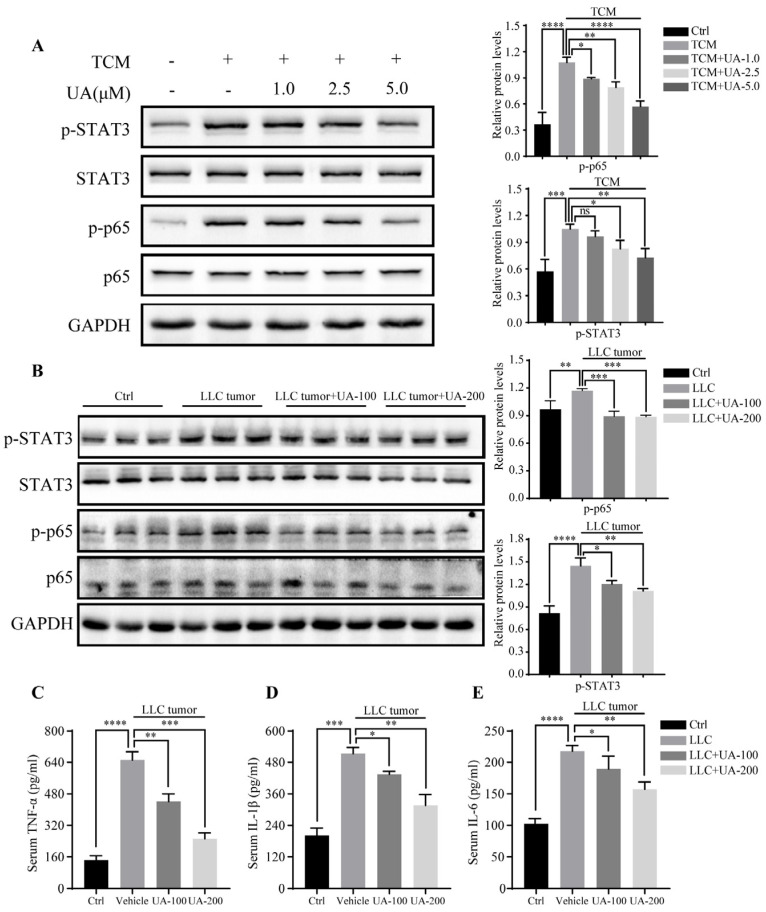
UA downregulates phosphorylation of STAT3 and p65 in vitro and in vivo and reverses elevated serum cytokines in the cancer cachectic model. The expression levels of indicated proteins (**A**) in the C2C12 myotube model and (**B**) in the gastrocnemius muscles of the animal model were determined by Western blot. The band intensities were quantified by densitometry, and the densities were quantified and normalized to GAPDH or the non-phosphorylated protein forms. The levels of pro-inflammatory cytokines (**C**) TNF-α, (**D**) IL-1β, and (**E**) IL-6 in the serum of the cancer cachectic model were detected by ELISA assay. * *p* < 0.05, ** *p* < 0.01, *** *p* < 0.001, **** *p* < 0.0001 versus control, ns: not significant, *n* = 3 in vitro and *n* = 10 mice/group in vivo.

**Figure 5 cancers-15-02378-f005:**
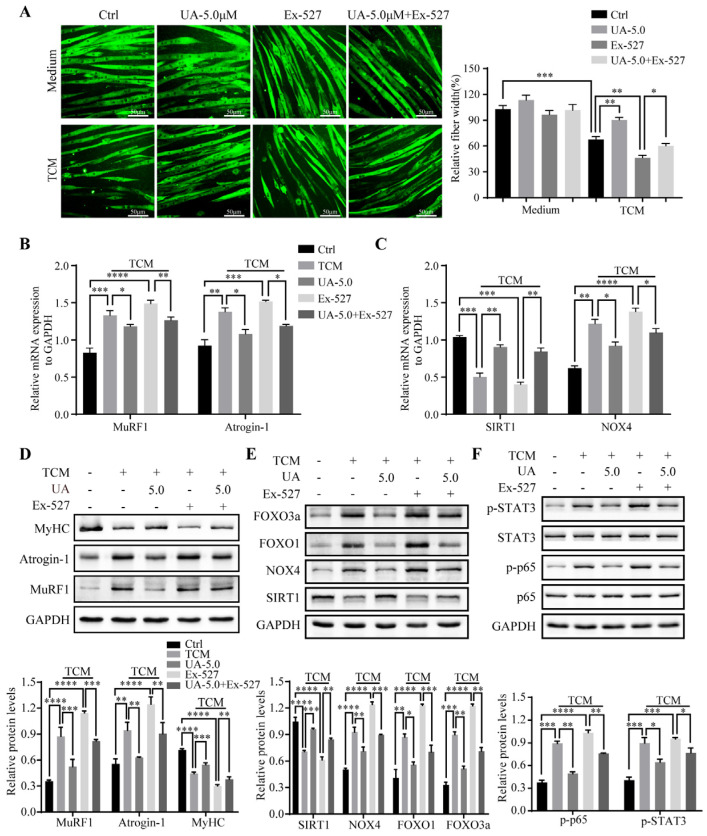
UA relieves TCM-induced myotube wasting through SIRT1 activation. C2C12 myotubes were pretreated with Ex-527 (0.1 μM) for 30 min and then stimulated by UA (5.0 μM) and TCM for 48 h. (**A**) Representative images of immunofluorescence staining for MyHC (green) are shown (left) with UA and Ex-527 treatment. Scale bars = 50 μm. The relative fiber widths of each experiment were measured and calculated (right). The mRNA expression levels of (**B**) MuRF1 and Atrogin-1 and (**C**) SIRT1 and NOX4 in the C2C12 myotube model were analyzed by real-time PCR and normalized to GAPDH. (**D**–**F**) Western blot was used to detect the expression of indicated proteins. The band intensities were quantified and normalized to GAPDH or the non-phosphorylated protein forms. * *p* < 0.05, ** *p* < 0.01, *** *p* < 0.001, **** *p* < 0.0001 versus control, ns: not significant, *n* = 3.

**Figure 6 cancers-15-02378-f006:**
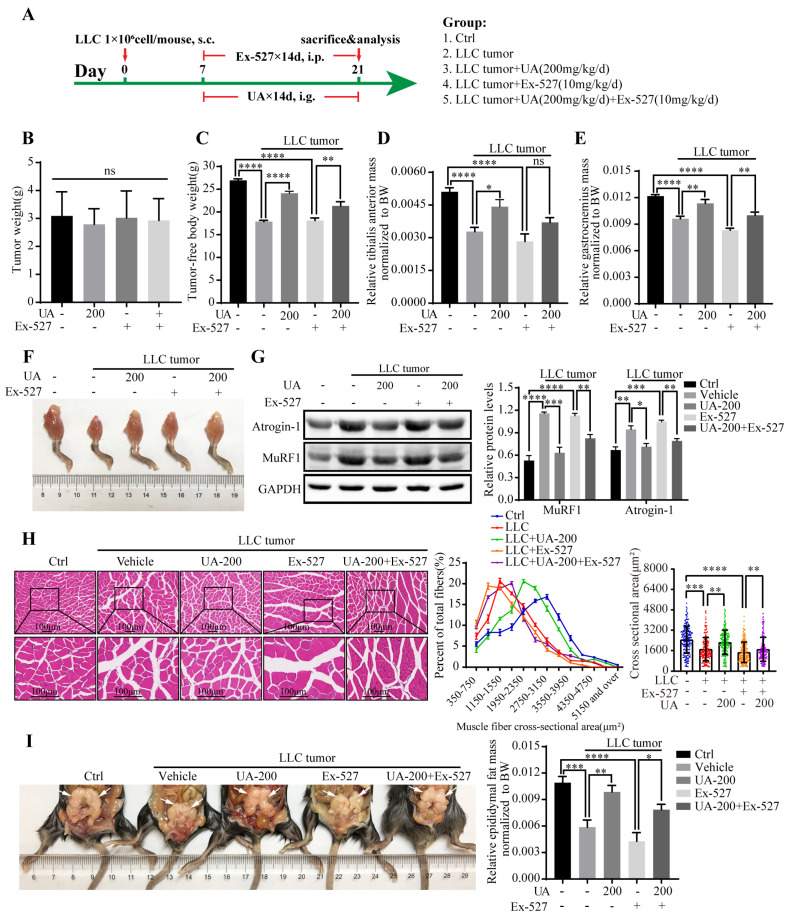
UA improves muscle wasting and prevents cancer cachexia in LLC tumor-bearing mice through SIRT1 activation. (**A**) Schematic illustration of the animal experimental design. The effects of UA on the main features of cachexia were examined, including (**B**) tumor weight, (**C**) tumor-free body weight, (**D**–**F**) tibialis anterior and gastrocnemius muscle mass, and (**I**) epididymal fat mass. (**G**) The expression of MuRF1 and Atrogin-1 in the gastrocnemius muscles of the cancer cachectic model was determined by Western blot. The band intensities were quantified by densitometry and normalized to GAPDH. (**H**) Gastrocnemius muscle was shown histologically by H&E staining. Scale bars = 100 μm. The cross-sectional areas were analyzed. * *p* < 0.05, ** *p* < 0.01, *** *p* < 0.001, **** *p* < 0.0001 versus control, ns: not significant, *n* = 10 mice/group.

**Figure 7 cancers-15-02378-f007:**
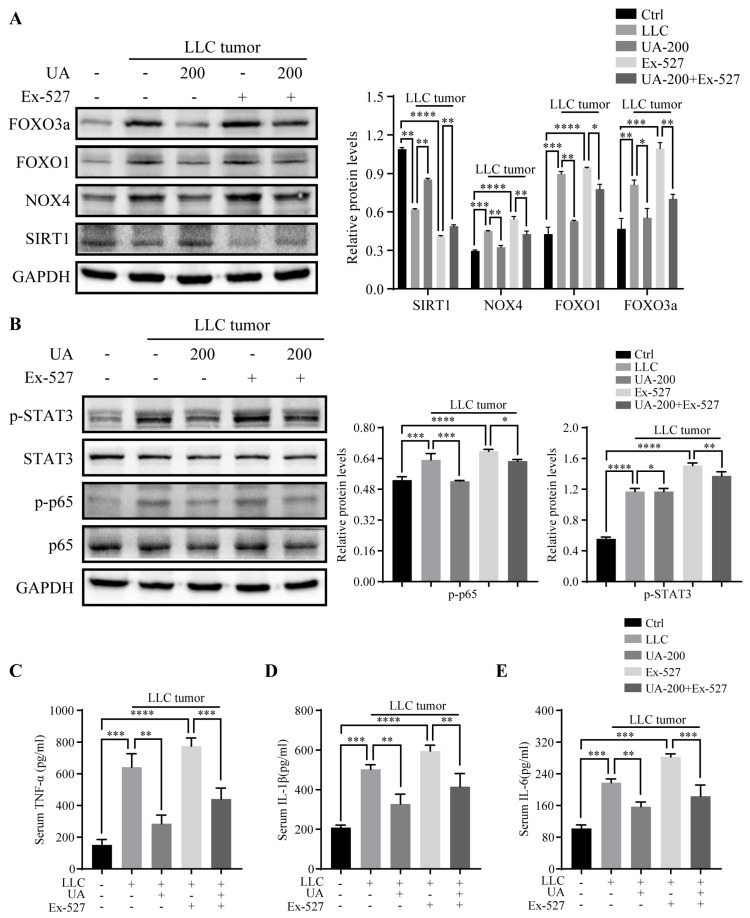
UA improves muscle wasting and prevents cancer cachexia in LLC tumor-bearing mice after Ex-527 (SIRT1 inhibitor) treatment. (**A**,**B**) Western blot was used to detect the expression of indicated proteins. The band intensities were quantified and normalized to GAPDH or the non-phosphorylated protein forms. The levels of pro-inflammatory cytokines (**C**) TNF-α, (**D**) IL-1β, and (**E**) IL-6 were detected by ELISA assay. * *p* < 0.05, ** *p* < 0.01, *** *p* < 0.001, **** *p* < 0.0001 versus control, *n* = 3.

**Figure 8 cancers-15-02378-f008:**
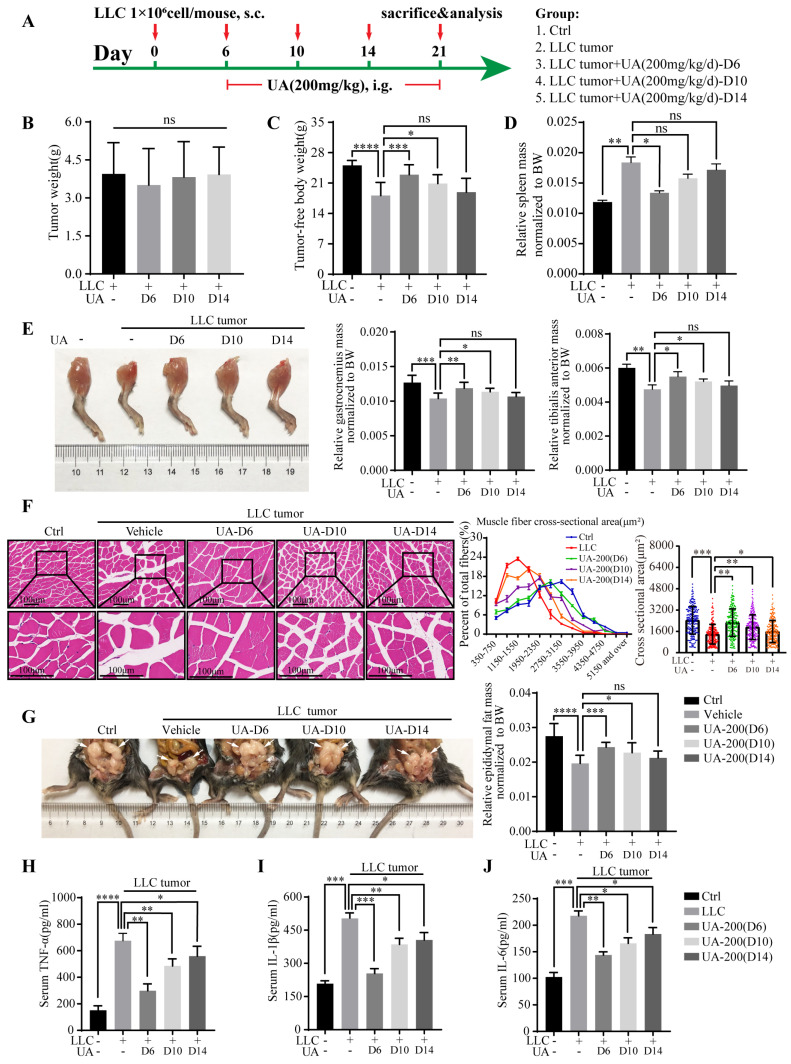
Effects of delaying treatment with UA until advanced stages of tumor and cachexia progression in LLC tumor-bearing mice. (**A**) Schematic illustration of the animal experimental design. The effects of UA on the main features of cachexia were examined, including (**B**) tumor weight, (**C**) tumor-free body weight, (**D**) spleen mass, (**E**) tibialis anterior and gastrocnemius muscle mass, and (**G**) epididymal fat mass. (**F**) Gastrocnemius muscle was shown histologically by H&E staining. Scale bars = 100 μm. The cross-sectional areas were analyzed. The levels of pro-inflammatory cytokines (**H**) TNF-α, (**I**) IL-1β, and (**J**) IL-6 in the serum of the cancer cachectic model were detected by ELISA assay. * *p* < 0.05, ** *p* < 0.01, *** *p* < 0.001, **** *p* < 0.0001 versus control, ns: not significant, *n* = 10 mice/group.

**Figure 9 cancers-15-02378-f009:**
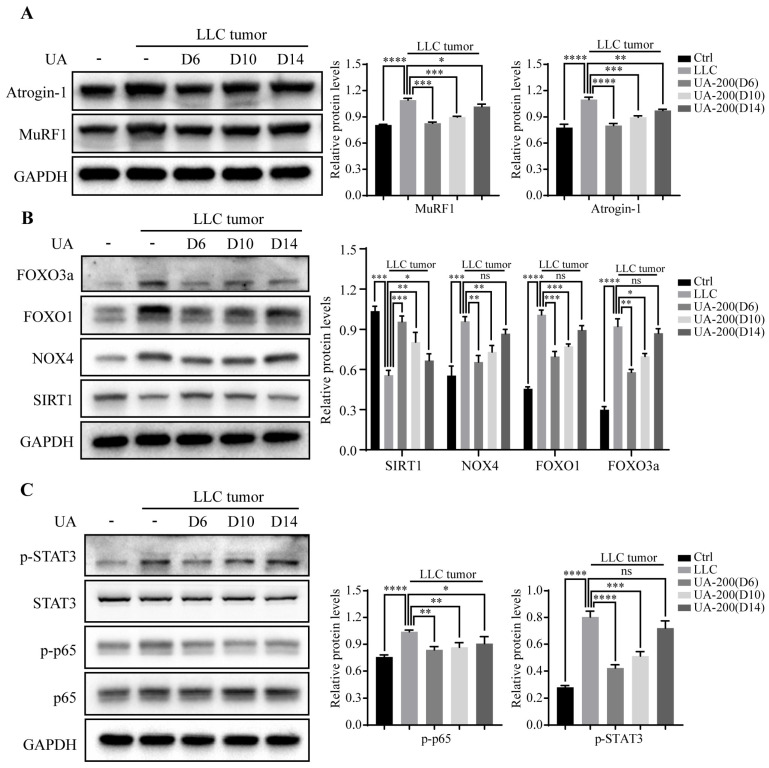
Effects of delaying treatment with UA until advanced stages of tumor and cachexia progression in LLC tumor-bearing mice on the expression of relevant molecules in the gastrocnemius muscle. (**A**–**C**) Western blot was used to detect the expression of indicated proteins. The band intensities were quantified and normalized to GAPDH or the non-phosphorylated protein forms. * *p* < 0.05, ** *p* < 0.01, *** *p* < 0.001, **** *p* < 0.0001 versus control, ns: not significant, *n* = 3.

## Data Availability

The data used to support the conclusions of this study will be made available by the corresponding author upon request.

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
