# Peer review of "Ursolic Acid Alleviates Cancer Cachexia and Prevents Muscle Wasting via Activating SIRT1"

_cancers, 2023, doi:10.3390/cancers15082378_

Round 1
Reviewer 1 Report
Comments to the authors
The manuscript by Tao et al. presents a study investigating the effects of ursolic acid on cancer cachexia. The authors used in vitro and in vivo models to demonstrate that ursolic acid has significant anti-cachectic effects. Furthermore, using this combination of in vitro and in vivo models, the authors provide a better understanding of the effects of this compound on biological processes. These results may lead to the development more effective and clinically relevant therapeutic interventions for the syndrome using this compound. Overall, the manuscript was well written and presented interesting results that provide valuable insight into this critical research area. However, the authors should address minor issues before publishing the manuscript. Thus, I have included some comments below to provide more specific feedback on the strengths and limitations of this submission.
Strengths
The results of the in vitro experiments suggest that ursolic acid can inhibit C2C12 myotube atrophy induced by Lewis lung carcinoma (LLC)–conditioned medium. In addition, the results of the in vivo experiments also support the efficacy of ursolic acid in suppressing cancer-induced muscle wasting. As the authors point out, these results are consistent with previous studies that reported that ursolic acid suppresses muscle wasting in other models.
Major comments for the authors to address before publication
In the abstract, the authors should include more information from the literature to help strengthen their hypothesis regarding the possible role of Ursolic acid in preventing muscle wasting. In particular, they should summarize the results of recent studies that demonstrated the beneficial effects of ursolic acid in animal models of muscle wasting.
It is recommended that all RT-qPCR experiments should be planned, conducted, and described according to the "MIQE Guidelines: Minimum Information for Publication of Quantitative Real-Time PCR Experiments" (Clinical Chemistry 55:4, 2009). Some important experimental details and results should be included in the present manuscript permitting a reader's ability to evaluate critically the quality of the qPCR results presented. Some significant points are described below:
- Lab-on-chip technology for automated capillary electrophoresis is state of the art and is recommended for standardized RNA integrity control. How did the authors check the RNA integrity?
- Reference genes should be selected depending on their apparent stability. How was determined the stability of the GAPDH (reference gene)? How does this normalization practice relate to the principles suggested by Vandesompele et al. in their papers on qPCR normalization to reference genes? Normalization against a single reference gene is inadequate unless the authors present clear evidence for the reviewers confirming its invariant expression under the experimental conditions described.
- How were the mRNA qPCR data normalized? According to the Methods section, the RT-qPCR data were normalized to GAPDH and expressed as relative mRNA expression using the delta-delta Ct. If the authors used the 2(-Delta Delta C(T)) Method described by Livak and Schmittgen, the reference should be included in the text.
- Please provide the primers and NCBI Reference Sequence for the six transcripts analyzed.
Authors may consider discussing the study results further and highlighting the significance of the findings beyond UA's proposed mechanism of action. This will provide a better understanding of the overall results of the study to the readers. Thus, please review the literature to include other processes that may be affected by the experimental manipulations. For example, the effect of UA on macrophages, tumor growth, and skeletal muscle satellite cells. Or, why did UA not change tumor growth as seen in other cancer cells or tumor types? The authors showed that UA had little effect on tumor growth, suggesting that it may have a selective anticancer effect.
These studies could provide the readers with a broader perspective of the effects of UA on the various cellular processes that were not investigated in the study. As highlighted by the authors, UA reduced levels of IL-6, IL-1β, and TNF-α in the serum of mice with LLC tumors and also inhibited the phosphorylation of NF-κB and STAT3. However, these effects were partially reversed by Ex-527.
Please check some examples below from the literature of other mechanisms involved in the in vitro and in vivo responses observed by the authors.
- Leng S, Hao Y, Du D, Xie S, Hong L, Gu H et al. Ursolic acid promotes cancer cell death by inducing Atg5-dependent autophagy. Int J Cancer 2013; 133: 2781–2790.
- Leng S, Iwanowycz S, Saaoud F, Wang J, Wang Y, Sergin I et al. Ursolic acid enhances macrophage autophagy and attenuates atherogenesis. J Lipid Res 2016; 57: 1006–1016.
- Zhao C, Yin S, Dong Y, Guo X, Fan L, Ye M, Hu H. Autophagy-dependent EIF2AK3 activation compromises ursolic acid-induced apoptosis through upregulation of MCL1 in MCF-7 human breast cancer cells. Autophagy 2013; 9(2):196-207.
- Katashima CK, Silva VR, Gomes TL, Pichard C, Pimentel GD. Ursolic acid and mechanisms of actions on adipose and muscle tissue: a systematic review. Molecules 2017; 22(11):1396.
- Bakhtiari N, Hosseinkhani S, Soleimani M, Hemmati R, Noori-Zadeh A, Javan M, Tashakor A. Short-term ursolic acid promotes skeletal muscle rejuvenation through enhancing of SIRT1 expression and satellite cells proliferation. Biomed Pharmacother. 2016; 78:185-196.
Minor comments for the authors to address before publication
There is scope to tighten up and refine the language to reflect the professional and scientific content. Perhaps it would be helpful to review and edit the manuscript to improve the clarity of the writing.
The authors should also revise the use of abbreviations. For example, in the section on the simple summary, UA is not used correctly.
In the graphical abstract, myocyte myofibers should be replaced by myofibers. Myocytes should be used when referring to the cardiac muscle cell (cardiomyocyte).
Please check the number of cells in line 138.
Line 259: typeset error for Atrogin-1
The legends in the x-axis of figure 8 are not very readable. Please remove some repetitive descriptions from the legend to help improve readability. For example, the redundant "200" could be omitted from Figures 8 b-j.
Reviewer 2 Report
This is a well written manuscript with an interesting mechanistic insight into the role of UA in muscle wasting due to cancer cachexia.
Specific Comments:
What is the active component in ursolic acid (UA)? Is there an understanding of the type of molecules it activates/binds to for mediating its effects? Seems like it may have off target or non-specific effects. Can the authors please comment on this? Figure 3A illustrates binding to a specific site in SIRT1 as described in lines 315-324, however can it be concluded/implied that this is SIRT1 specific? Demonstration of effects on other SIRTs would enhance this manuscript thereby suggesting that this family can be implicated in mediating UA effects.
Did the authors evaluate any other cell culture models i.e. cardiomyocyte? Since the effects of UA were on multiple tissues, an evaluation of another cell culture models that may emulate the cachectic phenotype of wasting would be useful to demonstrate UA effects. Therefore, a wider benefit of UA may be explored.
An additional mouse model such as C26 should be used in this report for comparative purposes. The molecular targets should be compared and contrasted across models of cancer cachexia as well as long and short cachexia progression i.e. over the longitudinal time course. This will demonstrate the effectiveness of UA across different settings. Soleus and EDL should also be included in this study.
Round 2
Reviewer 2 Report
Dear Authors, thank you for this revised manuscript.